# The Increase of the Magnitude of Spontaneous Viral Blips in Some Participants of Phase II Clinical Trial of Therapeutic Optimized HIV DNA Vaccine Candidate

**DOI:** 10.3390/vaccines7030092

**Published:** 2019-08-20

**Authors:** Ekaterina Akulova, Boris Murashev, Sergey Verevochkin, Alexey Masharsky, Ruslan Al-Shekhadat, Valeriy Poddubnyy, Olga Zozulya, Natalia Vostokova, Andrei P. Kozlov

**Affiliations:** 1Laboratory of Molecular Biology of HIV, Research Institute of Ultra Pure Biologicals, St. Petersburg 197110, Russia; 2The Biomedical Center, St. Petersburg 194044, Russia; 3Innovative Pharma, Mosco 143026, Russia

**Keywords:** HIV, AIDS, DNA vaccine, clinical trial, therapeutic vaccine

## Abstract

We developed a candidate DNA vaccine called “DNA-4”consisting of 4 plasmid DNAs encoding Nef, Gag, Pol(rt), and gp140 HIV-1 proteins. The vaccine was found to be safe and immunogenic in a phase I clinical trial. Here we present the results of a phase II clinical trial of “DNA-4”. This was a multicenter, double-blind, placebo-controlled clinical trial of safety, and dose selection of “DNA-4” in HIV-1 infected people receiving antiretroviral therapy (ART). Fifty-four patients were randomized into 3 groups (17 patients—group DNA-4 0.25 mg, 17 patients—group DNA-4 0.5 mg, 20 patients—the placebo group). All patients were immunized 4 times on days 0, 7, 11, and 15 followed by a 24-week follow-up period. “DNA-4” was found to be safe and well-tolerated at doses of 0.25 mg and 0.5 mg. We found that the amplitudes of the spontaneous viral load increases in three patients immunized with the candidate DNA vaccine were much higher than that in placebo group—2800, 180,000 and 709 copies/mL, suggesting a possible influence of therapeutic DNA vaccination on viral reservoirs in some patients on ART. We hypothesize that this influence was associated with the reactivation of proviral genomes.

## 1. Introduction

Since AIDS was first described in 1981 about 60 million people have been infected with HIV, and about 30 million have died of AIDS. In Russia more than 1.2 million infected individuals (59 per 100,000 of citizens) have been detected [1]. Despite significant progress having been made in the field of antiretroviral therapy (ART), the pandemic of HIV infection is yet to be contained. The development of vaccines against HIV/AIDS, both preventive and therapeutic, is the necessary step to stop further spread of the epidemic.

Our group has developed a candidate DNA vaccine called “DNA-4” which consists of 4 plasmid DNAs encoding Nef, Gag, Pol(rt), and gp140 HIV-1 proteins of the Eastern European subtype A. The candidate vaccine has passed preclinical studies in laboratory animals [2] and phase I clinical trials in healthy volunteers [3]. The vaccine was found to be safe and well-tolerated. Intramuscular immunization with “DNA-4” induced the development of HIV-specific mostly cellular immune responses in all trial participants. Some of the induced immune reactions, e.g., TNFα, were similar to the reactions discovered in exposed seronegative individuals, who remain HIV uninfected despite repeated unprotected exposure to the virus [3,4].

Here we present the results of a phase II clinical trial of the candidate vaccine “DNA-4” in HIV-1 infected people receiving ART. The objectives of the clinical trial were to assess safety and to determine an optimal dose of the vaccine for HIV-positive patients. We also were looking for the possible influence of vaccination on spontaneous increase of the viral load.

## 2. Materials and Methods

### 2.1. Study Vaccine

A candidate DNA-vaccine “DNA-4” has been developed at The Biomedical Center (St. Petersburg, Russia) in collaboration with the Research Institute of Ultra Pure Biologicals (St. Petersburg, Russia). The vaccine contains four plasmid DNA encoding consensus sequences of *nef*, *gag*, *rt*, or *gp140* HIV-1 FSU subtype A genes [2]. Amino acid sequences of viral proteins were modified to increase their expression level and optimize their immunological properties. Nucleotide sequences were designed to replace most wild-type codons with codons from highly expressed human genes. In reverse transcriptase (RT), N-terminal methionine and hystidines were introduced to replace catalytic aspartic acids residues 110, 185, and 186 within the active site of RT. In Nef, glycine residues 2 and 3 were deleted to remove the myristylation site. In gp140, the signal peptide was replaced with the signal sequence of human tissue plasminogen activator to increase its transport and secretion; the transmembrane and cytoplasmic regions of gp160 (amino acids 676–860) were removed to obtain a soluble form of the HIV-1 envelope glycoprotein, region 500–534 containing the cleavage site and fusion peptide domain was removed to prevent the proteolytic processing of the envelope, to stabilize the protein by linking it covalently to the gp41 extracellular domain, and to reduce toxicity; and region 589–618 containing the sequence between the heptad repeats was removed to stabilize the formation of trimers and eliminate formation of the hairpin intermediate [2].

Each gene was inserted into the vector pBMC that had been created at The Biomedical center. Inserted genes were expressed in eukaryotic cells under the control of the cytomegalovirus promoter and the bovine growth hormone polyadenylation signal [2].

DNA-4 was manufactured by the production facility of the Research Institute of Ultra Pure Biologicals (St. Petersburg, Russia) in accordance with the existing Russian federal regulations. The plasmids were equally formulated in 0.5 mL of sterile saline solution with overall plasmid concentration of 0.25 mg/mL. No adjuvants were added to the vaccine. Placebo vials contained 0.5 mL of saline solution without plasmids.

### 2.2. Phase II Clinical Trial Design

Phase II clinical trial was a multicenter, double-blind, placebo-controlled study. It was conducted to assess the safety of two “DNA-4” doses (0.5 mg and 0.25 mg) in patients with HIV-1 receiving ART by the analysis of frequency and severity of adverse events.

The study was conducted in 7 Centers for the Prevention and Control of AIDS and Infectious Diseases situated in different Russian cities: Moscow region, Kazan, Tolyatti, Volgograd, Lipetsk, Kaluga, Izhevsk. 

During screening (visit 1) the following data were obtained: medical history, assessment of weight and height, electrocardiography, chest X-ray (both direct and lateral projection), laboratory tests of blood and urine were performed, viral load, levels of CD4 and CD8 T cells. For women pregnancy tests were performed. Patients eligible for inclusion were included in the study. The inclusion and exclusion criteria used in the study are listed in Appendix A. All trial participants were randomized into three equal groups and vaccinated four times with corresponding dose (0.5 mg or 0.25 mg or placebo) on days 1, 7, 11, and 15 with a 22-week follow-up period. Vaccine doses were selected based on the results of the phase I clinical trials of DNA-4 vaccine [3]. The highest dose of 1.0 mg/mL was excluded from this study since it did not show the enhancement of the immunogenicity. 

Randomization was performed centrally by an unblinded study monitor according to the randomization list and stratum. At screening, each subject was allocated an individual registration. Investigator completed the Inclusion form including following information: screening date, site number, subject number, subject initials, date of birth, and basic ART. At Randomization visit the eligible patients were randomized to one of three treatment groups with the ratio 1:1:1. Trial participants were stratified by basic ART. Investigator indicated basic ART for each subject during randomization: 2NRTI + NNRTI or 2NRTI + PI. Patients with different basic ART were allocated equally to one of three treatment groups.

A dose of the studied vaccine was blinded by using two types of packages for each patient (box A and box B). Each package contained 4 ampoules with the DNA-4 vaccine with a dosage of 0.25 mg or with placebo.

Patients from 0.25 mg DNA-4 group were immunized with one ampoule from box A with DNA-4 vaccine of 0.25 mg intramuscularly strictly to the deltoid muscle of the right shoulder and one ampoule from box B with placebo intramuscularly to the deltoid muscle of the left shoulder.

Patients from 0.5 mg DNA-4 group were immunized with one ampoule from box A with DNA-4 vaccine of 0.25 mg intramuscularly strictly to the deltoid muscle of the right shoulder and one ampoule from box B with DNA-4 vaccine of 0.25 mg intramuscularly to the deltoid muscle of the left shoulder.

Patients from the placebo group were immunized with one ampoule from box A with placebo intramuscularly strictly to the deltoid muscle of the right shoulder and one ampoule from box B with placebo intramuscularly to the deltoid muscle of the left shoulder.

The candidate vaccine was administered intramuscularly in 1 mL of sterile saline solution in the deltoid muscle of each shoulder. Figure 1 shows the clinical trial design. 

Safety and tolerability were evaluated by the frequency and severity of adverse events (AE) according to subjective complaints from the patient’s diary, vital signs, physical examination, laboratory tests and development of local reactions. The severity of AE was assigned in accordance with the DAIDS scale, Version 1.0, December 2004. Each adverse event was graded using a 4-grade scale: 1—mild, 2—moderate, 3—severe, 4—potentially life threatening.

Association of AE with vaccine administration was determined as associated, possibly associated, unlikely associated, or not associated. AE associated with the vaccine injection should meet the following criteria: occurs in a short time after injection, accompanies a known response to the use of the vaccine, terminates after cessation of the vaccination, re-occurs after the resumption of the vaccination.

The viral load was assessed at screening and at visits 2 and 6–11 by real-time PCR analysis. “AmpliSense HIV-Monitor-M-FL” kit (Russia) were used to detect transient viral increases above 50 copies/mL (the sensitivity of the kit was 20 copies/mL). The magnitude of viral blips, the number of viral increases as well as the number of patients with viral increases were compared between vaccinated groups and placebo group.

The quantity and ratio of CD4 and CD8 T cells were measured by flow cytometry analysis.

The viral load and CD4 and CD8 T cell levels at visit 2 were the baselines for assessing the dynamics of the viral load.

### 2.3. Ethical Compliance

The study was reviewed and approved by the Ethical Committee of the Ministry of Health of the Russian Federation (clinical trial approval number 222 of 22 April 2014). The volunteers provided written informed consent following protocol review, as well as discussion and counseling with the clinical study team.

## 3. Results

### 3.1. Adherence and Tolerability

54 HIV-1 infected individuals receiving ART participated in the study. All participants were randomized into three equivalent groups: 0.5 mg of vaccine—17 individuals, 0.25 mg—17 individuals, and placebo—20 individuals. Demographic characteristics of trial participants are presented in Table 1.

Vaccination was fully completed in 53 trial participants. One individual from group vaccinated with 0.25 mg of the vaccine was prematurely withdrawn from the study after first vaccine application due to a cold caused by a respiratory virus. There was no temporal association with vaccine administration, so this AE was determined as unlikely to be associated with the vaccination. However, data on safety and tolerability were analyzed in this participant. 

The diagram describing the course of the study is presented at Figure 2.

Adverse events were registered in 17 out of 54 trial participants (31.8%). In the vaccinated groups (0.25 mg and 0.5 mg combined) 35 AE in 12 patients were detected (35.3%), in the placebo group—13 AE in 5 patients (25.0%). In the group receiving 0.25 mg of the vaccine AE were found twice as often as in the group receiving the 0.5 mg dose (47.1% and 23.5% respectively). The total data on the adverse events registered in trial participants are presented in Table 2. Statistically significant differences between the frequencies of adverse events in vaccinated and placebo groups were not found (Fisher’s exact test).

Pain in the left arm and hyperemia at the injection site were associated with immunization with the studied candidate vaccine. Fourteen cases of AE were determined to be possibly associated with vaccination including leukopenia, neutropenia, fever, itching at the injection site, hypersecretion from the genital tract, and menstrual disorders.

No deaths were detected. Most adverse events had mild or moderate severity. In the vaccinated groups, 4 cases of 3rd grade AE (3 cases in the 0.25 mg group and 1 case in the 0.5 mg group) and 1 case of 4th grade AE (in the 0.25 mg group) were registered, all of them neutropenias. This did not lead to an interruption of the vaccination. In all cases neutropenias were completely resolved.

### 3.2. Viral Load Dynamics

A viral load was measured at screening and at visits 2 and 6–11 by real-time PCR analysis. Transient viral increases above 50 copies/mL (viral blips) were analyzed. Table 3 presents the data on viral load analysis.

The relative frequency of the viral blips as well as the number of patients with viral blips in the placebo group and the vaccinated groups did not differ (Table 3). But the magnitude of some transient viral increases was much higher in groups receiving the candidate DNA vaccine. The biggest blips were detected in the group receiving 0.25 mg of the vaccine (patients # 21 and 37) and made up 2800 and 18,000 copies/mL, respectively. There were gradual increases and then decreases of the viral load in participant #21. The third largest blip, 709 copies/mL, was found in 0.5 mg group in patient #43 (Appendix B
Table A1).

### 3.3. CD4 and CD8 T Cells Measurement

The number of CD4 and CD8 T cells were measured by flow cytometry at visits 2, 6, 7, 9, and 11. At visit 2 blood donation was performed before the first vaccination. Results at visit 2 present the data on the CD4 and CD8 level at the time of study entry. The results are shown in Table 4 and Table 5.

There was a weak trend to increase in the absolute number of CD4 T cells in the group receiving 0.25 mg of the studied vaccine, but the differences with the placebo group were not statistically significant.

## 4. Discussion

Different therapeutic vaccine strategies including tools based on DNA, viral vectors such as modified vaccinia Ankara (MVA) and vesicular stomatitis virus (VSV), RNA, peptide, or protein, Lentiviral vector and dendritic cell have been used in numerous clinical trials. Despite the major advances in our immunological understanding of HIV-1 specific T cell responses and HIV-1 reservoir, we have not been able to achieve a cure and none of these vaccines have proven to be effective [5]. Combination strategies are now being considered as the most promising approach for therapeutic HIV vaccine development. Interleukins, immune checkpoint inhibitors and Treg modulation were suggested as candidates for effective vaccine, but failed to yield any significant clinical benefit [5]. At the CROI 2017 conference data on clinical trial BCN02 were presented. It was the combined use of therapeutic vaccination with a vaccine based on the MVA vector (MVA.HIVconsv vaccine) and Romidepsin, specific drugs that can reactivate latent virus from the reservoir (Kick and kill strategies) followed by ART treatment interruption. At the time of the report 11 patients had interrupted treatment, 7 of them had to resume ART within the first 4 weeks while 4 participants (36%) remained off ART after 7, 12, 14, and 22 weeks, respectively. The authors suggested that therapeutic vaccination targeting conserved regions of HIV-1 combined with HIV latency reactivation strategies may facilitate clearance of the viral reservoir in early-treated individuals [6]. The DNA vaccine, since it induces HIV specific cytotoxic T cells, in case of latent viral reservoirs destruction may be an ideal strategy for HIV eradication.

In our previous studies we have developed a candidate DNA vaccine against HIV-1 consisting of four plasmids encoding four HIV-1 subtype A genes: *gag*, *env*, *rt*, and *nef* [2]. The preclinical studies and phase I clinical trial of the vaccine were conducted [2,3]. The phase I trial was conducted to access safety, tolerability and immunogenicity of the DNA-4 HIV vaccine in healthy HIV-1-negative adult volunteers. We found that our DNA vaccine was safe and well-tolerated at three used doses (0.25 mg, 0.5 mg, and 1.0 mg). Altogether, T-cell immune responses were elicited in all participants. We observed the increase in lymphocyte proliferation after fourth immunization that can show the advantage of fourfold against triple immunization. The frequency of detection positive cytokine responses decreases with increasing the vaccine dose. The humoral responses were induced in 5 people (24%). We did not observe any correlation between the antibody production and the DNA-4 vaccine doses. We also found the important correlation with our results obtained for the HIV specific immune responses in exposed seronegative individuals, i.e., TNFa production in immunized group [3].

This study was conducted as a multicenter, double-blind, placebo-controlled study of safety and dose selection of a candidate HIV vaccine for HIV-infected people receiving ART. It can be concluded that the DNA-4 candidate vaccine at doses of 0.25 mg and 0.5 mg was safe and well-tolerated by HIV-infected individuals receiving ART. In vaccinated groups, three spontaneous increases of viral load with largest amplitude were detected.

The proportion of trial participants who demonstrated adverse events associated or possibly associated with the vaccine administration was 7.4% higher in the vaccinated group than in the placebo group. The frequency of local reactions in group immunized with 0.25 mg of the vaccine and the placebo group was similar, and in the group immunized with 0.5 mg no local reactions were revealed. This is in contrast with other AE, which were highest in the group receiving 0.25 mg of the vaccine. 

Immunogenicity of the DNA-4 vaccine was performed in Phase I clinical trial using IFNγ-ELISpot, intracellular cytokine staining (ICS) of IFNγ, TNFα and IL-2, lymphocyte proliferation assay (LPA) and ELISA [3]. For specific T cell stimulation, a panel of 451 overlapping peptides spanning HIV-1 subtype A-Eastern European (EE) Env, Gag, RT, and Nef proteins was used. Peptides were synthesized at the Research Institute of Ultra Pure Biologicals (St. Petersburg, Russia). HIV-specific cellular immune responses were detected in 21/21 (100%) trial participants: 9 patients were IFNγ-ELISpot reactive, 18 patients expressed cytokines to specific antigen stimulation, and 12 patients had positive lymphocyte proliferation. Using ICS we detected the increased TNFα expression by CD4 T cells in response to the specific peptide stimulation in 3/21 trial participants [3]. The humoral response was induced in 5 people (24%). The titer of HIV-specific antibodies did not exceed 1/100.

For complete eradication of the HIV infection the destruction of latent viral reservoirs is necessary, and this cannot be achieved by modern ART. The only example of HIV cure is the so-called “Berlin patient” who underwent allogeneic hematopoietic stem-cell transplantation (HSCT) from a donor carrying homozygous mutation in the HIV coreceptor CCR5 [7,8]. Recently information about HIV-1 remission maintained over a further 18 months after a similar procedure has been published [9]. However, this procedure is very expensive, high-risk and cannot be widely used.

One of the approaches used for eliminating viral reservoirs is reactivation of latent proviral genomes during ART treatment by histone deacetylase inhibitors and some cytokines [10,11].

Another way is enhancement of cellular immunity in HIV-infected individuals using therapeutic vaccines capable of inducing functional CD8^+^ T cells specific for HIV-1 epitopes [12,13,14,15,16]. The next generation of therapeutic vaccines will also be combined with reservoir activating agents [17]. DNA vaccines, in case of provirus activation, may be an ideal drug for viral reservoirs eradication.

Proviral genome reactivation may be caused by TNFα expression. TNFα activates transcription factor NFkB and HIV transcription [12,18]. DNA-4 vaccination induced increased TNFα expression in some individuals, as shown in a phase I clinical trial by intracellular cytokine staining followed by flow cytometry [3]. The expression of TNFα was also demonstrated by us in a cohort of exposed, seronegative individuals [4]. That is why we hypothesized that therapeutic “DNA-4” vaccine immunization may activate latent provirus and destroy at least some virus reservoirs. In order to measure that, we assessed the frequency and magnitude of transient viral load increases above 50 copies/mL (blips). Such spontaneous viral load increases occurring during ART treatment may be associated with latent viral reservoirs activation. 

To investigate the possible effects of DNA vaccination on viral reservoirs we analyzed the magnitude and frequency of the blips in the placebo and immunized groups (Table 3). Neither relative frequency of the blips nor relative numbers of patients with blips differ between groups. But the amplitudes of blips in patients 21 and 37 immunized with 0.25 mg of the candidate DNA vaccine were much higher than that in placebo group—2800 and 18,000 copies/mL, respectively. In participant #21 an increase of the viral load was detected from the 6th to the 10th visit with the dynamics of increasing, peak and decreasing of the viral load, while most of the other blips were detected only during a single visit. The third largest blip, 709 copies/mL, was found in trial participant 43 vaccinated with 0.5 mg of DNA vaccine.

The number of patients is small. However, the largest increases were registered in double-blinded vaccinated groups. The results suggest that the lower DNA concentration (0.25 mg) is more active than 0.5 mg. This is in correspondence with more AE in the group vaccinated with 0.25 mg.

These results may have several explanations. The participation of therapeutic DNA-4 vaccination during ART in destruction of latent viral reservoirs in some patients due to the reactivation of a latent provirus by TNFα is possible but is not proved. The destruction of latent cells containing viral RNA can be the source of the viral blips.“Repliclones”, populations of replicating cells with HIV’s genome nested inside them can also produce new virions [19]. 

The studied vaccine contains Nef protein which has been shown to have an ability to induce viral reactivation. It was demonstrated that exogenous Nef activated virus production in latent cell lines and in peripheral blood mononuclear cells isolated from asymptomatic HIV-infected individuals [20]. Early production of Nef during viral reactivation might enhance latent T cell activation.

Nef increases the production of exosomes containing activated ADAM17 (a disintegrin and metalloprotease domain 17), an enzyme that converts pro-TNF-α into its active form. The uptake of ADAM17-containing exosomes by target cells can induce the release of TNF-α, which subsequently binds to TNF receptor type 1 and activates NF-κB and c-Jun N-terminal kinase (JNK) pathways [20].

On the other hand, Nef is able to selectively downregulate surface CD4 and HLA-I molecules that may lead to evade immune surveillance by reactivated cells. Moreover, Nef can counteract multiple apoptotic pathways and promote cell survival could further hinder the clearance of reactivating reservoirs [20]. So, Nef protein can has dual effect on latent viral reservoirs reactivation.

## 5. Conclusions

The further studies of these effects are necessary. As far as we know, no systematic studies of blips in vaccinated patients have been performed before. The measurement of the magnitude of spontaneous increases of viral load could become the part of monitoring the results of immunotherapy of HIV-infected patients in the future.

In conclusion, we demonstrated safety of the candidate DNA vaccine in HIV-infected patients receiving ART, and detected unusual blips effects in vaccinated individuals, which may be of interest for the future studies.

## Figures and Tables

**Figure 1 vaccines-07-00092-f001:**
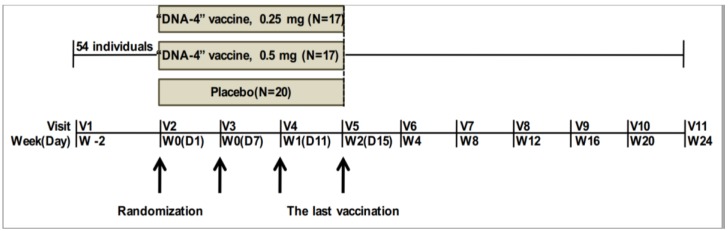
Trial scheme. Arrows show days of immunization.

**Figure 2 vaccines-07-00092-f002:**
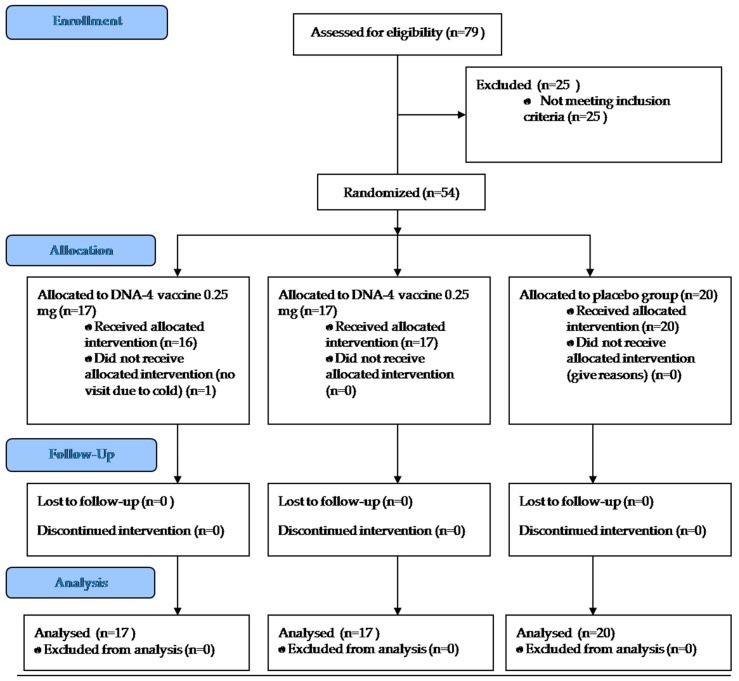
Consort diagram of the study.

**Table 1 vaccines-07-00092-t001:** Demographic characteristics of trial participants.

Group	Placebo	0.25 mg	0.5 mg	Total
Number of participants randomized	20	17	17	54
Number of men	9	5	9	23
Number of women	11	12	8	31
Average age	33.6 ± 6.2	36.9 ± 9.4	37.1 ± 8.5	35.7 ± 8.0

**Table 2 vaccines-07-00092-t002:** Adverse events registered in trial participants.

Adverse Event	DNA-4 0.25 mg		DNA-4 0.5 mg		Placebo	
Number	17		17		20	
	N	%	N	%	N	%
Fever	1	5.9	0	0.0	2	10.0
Feeling of acid in the mouth	0	0.0	0	0.0	1	5.0
Toothache	0	0.0	0	0.0	1	5.0
Weakness	0	0.0	0	0.0	1	5.0
Left arm pain	0	0.0	0	0.0	1	5.0
Itching at the injection site	1	5.9	0	0.0	1	5.0
Hyperemia at the injection site	1	5.9	0	0.0	0	0.0
Menstrual disorders	2	18.2	0	0.0	0	0.0
Hypersecretion from the genital tract (subjective analysis)	1	5.9	0	0.0	0	0.0
Cold	3	17.6	3	17.6	0	0.0
Gastrointestinal infection	1	5.9	0	0.0	0	0.0
High blood pressure	1	5.9	0	0.0	0	0.0
Neutropenia	3	17.6	1	5.9	1	5.0
Increased bilirubin	1	5.9	0	0.0	1	5.0
Leukopenia	2	11.8	2	11.8	1	5.0
Anemia	1	5.9	0	0.0	1	5.0
Increase in alanine aminotransferase	1	5.9	0	0.0	0	0.0
Increase in gamma-glutamyl transferase	1	5.9	0	0.0	0	0.0
Erythropenia	0	0.0	1	5.9	0	0.0
Proteinuria	0	0.0	1	5.9	0	0.0
Irritability	0	0.0	1	5.9	0	0.0

*p* > 0.05.

**Table 3 vaccines-07-00092-t003:** Data on viral load increases registered in trial participants.

Group	0.25 mg		0.5 mg		Placebo	
The number of viral blips (>50 copies/mL)	8/88	9.1%	6/89	6.7%	7/95	7.4%
The number of participants with viral blips	4/17	23.5%	6/17	35.3%	6/20	30.0%

*p* > 0.1.

**Table 4 vaccines-07-00092-t004:** Data on CD4 T cells number in trial participants at different visits (cells × 10^9^/L).

Group		Visit						
		Screening	2	6	7	9	11	*t* test
0.25 mg	N	17	16	17	15	15	13	0.132
	Mean	0.669	0.593	0.619	0.722	0.703	0.756	
	SD	0.224	0.223	0.241	0.152	0.223	0.187	
	Min	0.289	0.246	0.288	0.506	0.354	0.348	
	Max	1.086	1.114	1.290	1.176	1.121	1.056	
0.5 mg	N	17	17	17	14	14	11	0.104
	Mean	0.707	0.769	0.714	0.710	0.671	0.797	
	SD	0.259	0.282	0.278	0.278	0.255	0.323	
	Min	0.289	0.307	0.333	0.346	0.232	0.361	
	Max	1.157	1.281	1.267	1.275	1.093	1.196	
Placebo	N	20	20	20	17	16	12	-
	Mean	0.567	0.558	0.620	0.603	0.555	0.650	
	SD	0.197	0.202	0.183	0.213	0.127	0.211	
	Min	0.336	0.189	0.353	0.278	0.239	0.367	
	Max	1.159	1.063	1.018	1.009	0.749	1.009	

**Table 5 vaccines-07-00092-t005:** Data on CD8 T cells number in trial participants at different visits (cells × 10^9^/L).

Group		Visit						
		Screening	2	6	7	9	11	*t* test
0.25 mg	N	17	16	17	15	15	13	0.306
	Mean	0.926	0.939	0.937	1.059	1.014	0.937	
	SD	0.413	0.540	0.450	0.533	0.534	0.293	
	Min	0.347	0.312	0.316	0.301	0.390	0.352	
	Max	1.776	2.376	1.926	2.199	2.230	1.406	
0.5 mg	N	17	17	17	14	14	11	0.969
	Mean	1.023	1.037	0.992	0.902	0.847	1.033	
	SD	0.477	0.479	0.380	0.276	0.390	0.451	
	Min	0.358	0.422	0.322	0.507	0.262	0.469	
	Max	1.871	2.020	1.870	1.287	1.635	2.033	
Placebo	N	20	20	20	17	16	12	-
	Mean	0.975	0.976	0.976	0.940	0.923	1.086	
	SD	0.436	0.549	0.484	0.408	0.349	0.500	
	Min	0.299	0.285	0.332	0.364	0.437	0.476	
	Max	2.169	2.506	2.059	1.711	1.604	2.365	

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
