# Peer review of "The Increase of the Magnitude of Spontaneous Viral Blips in Some Participants of Phase II Clinical Trial of Therapeutic Optimized HIV DNA Vaccine Candidate"

_vaccines, 2019, doi:10.3390/vaccines7030092_

Round 1

Reviewer 1 Report

This is a manuscript describing the results of what is described as a Phase II trial of plasmid DNA vaccine encoding different HIV proteins.  This is very important work by a highly respected team of Russian scientists.   The trial has several strengths, including being a multicenter randomized, placebo controlled trial. It is not clear however, why this is a Phase II trial- what are the efficacy outcomes that are being sought that would provide evidence for moving on to a Phase III trial?  Certainly the vaccine seems safe enough, although there is a concern about neutropenia. However the viral blips described in 3 participants seems like weak evidence for efficacy.  What have the investigators concluding from the dosing- both doses are safe but there was no evidence for higher “efficacy” in the higher dose group.  Although the statement by the investigators that “therapeutic DNA-4 vaccination during ART may cause the destruction of latent viral reservoirs in some patients due to reactivation of latent provirus” is certainly possible, the data presented in this manuscript do not strongly support it. An additional weakness is that there are a paucity of data on the participants, including their adherence to ART during the study, inclusion and exclusion criteria, entry CD4 level and HIV viral load. No data are presented on immunogenicity.  Randomization may have been a problem given the skewed sex ratio in one of the groups. Finally, It would be helpful for the investigators in the discussion to summarize what they have learned from this trial that would allow them to move to the next stage of investigation of therapeutic HIV vaccination.

Abstract

Line 14- should read “a “ phase I clinical trial

Line 15- should read “a “ phase II clinical trial

Line 17- would be helpful to specify if the patients on ART were required to have an undetectable HIV viral load at the time of starting vaccination.

Line 18- should read “patients”

Line 19- should read “patients” and “placebo group”

Line 20- should read “a 24-week”

Line 21- rather than saying that they obtained data, the authors should say what those data were. Otherwise it is difficult to assess the accuracy of the conclusions.

Introduction

Line 32- should read “a candidate DNA vaccine”

Line 33- should read “the Eastern”

Line 34- should read “clinical trials”

Line 35- should read “Intramuscular immunization…”

Line 37- should read “Some of the”

Line 40- should read “results of a”

Line 42- should read “patients”

Line 44- should specify if the increase in magnitude was compared with the placebo group.  Also was it just magnitude, or increase in the number of times HIV was detected or both? 

Line 47- Did the vaccine contain an adjuvant?  Details of the plasmids are given in reference 2 but readers of this article may not have easy access to that article- it might be useful to include these details in a table in a figure in this paper, or perhaps an appendix. 

Methods

Line 48- should read “the Research Institute”

Line 61- should read “participants were randomized”

Line 63- participants were injected 4 times- was that done by alternating shoulders?  What was done at visit 1?  What were the inclusion and exclusion criteria for the study?

Line 67- how were subjective complaints assessed- through a diary?

Line 70- should read “Adverse events were graded using a 4-grade..”

Results

More demographic information is needed.  Table 1- How was randomization actually performed?  There appears to be a problem with randomization since the 0.25 mg group has a lower male:female ratio than the other groups. The placebo group also appears to be younger than the vaccine groups. How did the age of the men compare with the age of the women? What was their HIV risk group?  How did they compare with respect to CD4 nadir, current CD4 at the time of entering the study and length of time on ART?  How was it determined that participants were taking their ART exactly as prescribed during the study? It would be helpful to indicate where there are statistically significant differences between groups.

Line 85- should read “are presented”

Line 87- which arm was the person who withdrew randomized to? After how many injections? Other than this one person, did all other participants receive al 4 injections?

Table 2- it would be helpful to indicate where there are statistically significant differences between groups. This reviewer suggests that the column combining the 2 vaccine groups be removed as it is not helpful. Likewise for information combining the 2 groups in the text- data should only be presented separately for each group. It is surprising that some AE such as local injection site reactions are more common in the lower dose vaccine group than the higher dose group. What is meant by “hypersecretion of the genital tract”?  For menstrual disorders, the proportion should be expressed as a proportion of the women in the group, not the entire group. What is meant by food infection?  How severe was the neutropenia?  What is meant by leukopenia?  Precise definitions of what is meant by increases or decreases in the different laboratory measures must be provided. 

Line 91- should read “In the group…”

Line 95- the first sentence in this paragraph is not clear. 

Line 99- which groups had the grade 3 and grade 4 AEs?

Table 3- The authors should state what the numbers in the table represent.  When the number 20 is shown, was that the actual copy number detected or does that mean that the VL was undetectable in this assay?  The group data are very difficult to understand in this table.

Line 108- should read “magnitude”

Line 111- should read “gradual increases and then decreases”

Table 4- the table should include the CD4 level at the time of study entry. The units of the results should be given- e.g., cells/microliter, etc? 

Was there any relationship between VL at the time of study entry and detection of VL above 50 after study entry? Was there any relationship between CD4 level  at the time of study entry and detection of change in CD4 level or viral load after study entry? 

In the abstract the authors describe this as a trial of immunogenicity, but there are no immunogenicity data presented in this manuscript.

Discussion

It would be good to have more discussion in the background about other therapeutic HIV vaccines and how they compare with the candidate vaccine described in this manuscript. 

Line 122- should read “This study

Line 122- it is not clear what criteria were going to be used to select a dose.  In some ways this trial represents a Phase I dose-finding study more than a Phase II trial where the study was not designed to look at efficacy. 

Line 129- this information belongs in the results, not the discussion. 

Line 132- before it can concluded that the vaccine is safe, more information is needed about the neutropenia- did it resolve?  How extensive was it? Why was it seen more in the lower dose vaccine group?

Line 137- should read “Recently information..”

Line 138- should read “after a similar procedure..”

Line 145- should read “vaccines…”

Line 148- should read “as shown in a phase I…”

Line 149- should read “in a cohort”

Line 154- as the authors indicate, there are several possibleexplanations of blips appearing.  No real evidence that this was due to reactivation of the latent viral reservoir was presented.  As the authors indicated, neither relative frequency of the blips nor relative numbers of patients with blips differed between groups.  The only evidence presented were sizeable blips in 2 patients in the lower dose vaccine group, and a smaller one in one patient in the larger dose group.  Do we have information on consistent use of ART by these patients? Might there have been any other concurrent events that might have explained those blips, e.g., a cold or other infection?

Author Response

Dear Reviewer,

thank you for reviewing the manuscript. Please, find attached a point-by-point response to your comments.

Point 1: Line 14- should read “a “ phase I clinical trial 

Response 1: corrected

Point 2: Line 15- should read “a “ phase II clinical trial

Response 2: corrected

Point 3: Line 17- would be helpful to specify if the patients on ART were required to have an undetectable HIV viral load at the time of starting vaccination.

Response 3: the information has been added

Point 4: Line 18- should read “patients”

Response 4: corrected

Point 5: Line 19- should read “patients” and “placebo group”

Response 5: corrected

Point 6: Line 20- should read “a 24-week”

Response 6: corrected

Point 7: Line 21- rather than saying that they obtained data, the authors should say what those data were. Otherwise it is difficult to assess the accuracy of the conclusions.

Response 7: the information has been added

Point 8: Line 32- should read “a candidate DNA vaccine”

Response 8: corrected

Point 9: Line 33- should read “the Eastern”

Response 9: corrected

Point 10: Line 34- should read “clinical trials”

Response 10: corrected

Point 11: Line 35- should read “Intramuscular immunization…”

Response 11: corrected

Point 12: Line 37- should read “Some of the”

Response 12: corrected

Point 13: Line 40- should read “results of a”

Response 13: corrected

Point 14: Line 42- should read “patients”

Response 14: corrected

Point 15: Line 44- (Materials and Methods) should specify if the increase in magnitude was compared with the placebo group.  Also was it just magnitude, or increase in the number of times HIV was detected or both? 

Response 15: the information has been added

Point 16: Line 47- Did the vaccine contain an adjuvant?  Details of the plasmids are given in reference 2 but readers of this article may not have easy access to that article- it might be useful to include these details in a table in a figure in this paper, or perhaps an appendix. 

Response 16: the information has been added

Point 17: Line 48- should read “the Research Institute”

Response 17: corrected

Point 18: Line 61- should read “participants were randomized”

Response 18: corrected

Point 19: Line 63- participants were injected 4 times- was that done by alternating shoulders?  What was done at visit 1?  What were the inclusion and exclusion criteria for the study?

Response 19: the information has been added

Point 20: Line 67- how were subjective complaints assessed- through a diary?

Response 20: the information has been added

Point 21: Line 70- should read “Adverse events were graded using a 4-grade..”

Response 21: corrected

Point 22: Table 1- How was randomization actually performed?  There appears to be a problem with randomization since the 0.25 mg group has a lower male:female ratio than the other groups. The placebo group also appears to be younger than the vaccine groups. How did the age of the men compare with the age of the women? What was their HIV risk group?  How did they compare with respect to CD4 nadir, current CD4 at the time of entering the study and length of time on ART?  How was it determined that participants were taking their ART exactly as prescribed during the study? It would be helpful to indicate where there are statistically significant differences between groups.

Response 22: the information about randomization process has added. Randomization was performed centrally by an unblinded study monitor according to the randomization list and stratum. Trial participants were stratified by basic ART. The viral load, number of CD4 T cells, length of time on ART were according to inclusion criteria. The adherence to ART was assessed by self-reported data.

Point 23: Line 85- should read “are presented”

Response 23: corrected

Point 24: Line 87- which arm was the person who withdrew randomized to? After how many injections? Other than this one person, did all other participants receive al 4 injections?

Response 24: the information has been added

Point 25: Table 2- it would be helpful to indicate where there are statistically significant differences between groups. This reviewer suggests that the column combining the 2 vaccine groups be removed as it is not helpful. Likewise for information combining the 2 groups in the text- data should only be presented separately for each group. It is surprising that some AE such as local injection site reactions are more common in the lower dose vaccine group than the higher dose group. What is meant by “hypersecretion of the genital tract”?  For menstrual disorders, the proportion should be expressed as a proportion of the women in the group, not the entire group. What is meant by food infection?  How severe was the neutropenia?  What is meant by neutropenia?  Precise definitions of what is meant by increases or decreases in the different laboratory measures must be provided. 

Response 25: the column removed. The information about adverse events has added. The increases and decreases in the laboratory measures were determined according to standards used in Russian hospitals.

Point 26: Line 91- should read “In the group…”

Response 26: corrected

Point 27: Line 95- the first sentence in this paragraph is not clear. 

Response 27: corrected

Point 28: Line 99- which groups had the grade 3 and grade 4 AEs?

Response 28: the information has been added

Point 29:  Table 3- The authors should state what the numbers in the table represent.  When the number 20 is shown, was that the actual copy number detected or does that mean that the VL was undetectable in this assay?  The group data are very difficult to understand in this table.

Response 29: the information has been added

Point 30: Line 108- should read “magnitude”

Response 30: corrected

Point 31: Line 111- should read “gradual increases and then decreases”

Response 31: corrected

Point 32: Table 4- the table should include the CD4 level at the time of study entry. The units of the results should be given- e.g., cells/microliter, etc? 

Response 32: the information has been added

Point 33: Was there any relationship between VL at the time of study entry and detection of VL above 50 after study entry? Was there any relationship between CD4 level  at the time of study entry and detection of change in CD4 level or viral load after study entry? 

Response 33: The inclusion criteria included the demand that viral load should be less than 50 copies/ml and number of CD4 T cells more than 250 cells/mm3 at screening. No correlation was found.

Point 34: In the abstract the authors describe this as a trial of immunogenicity, but there are no immunogenicity data presented in this manuscript

Response 34: Immunogenicity of the vaccine was studied in Phase I clinical trial by IFNγ-ELISpot, intracellular cytokine staining of IFNγ, TNFα, and IL-2, lymphocyte proliferation assay and ELISA. In Phase II clinical trial we assessed just common immunological parameters of trial participants to study the influence of the vaccination on immune system of patients.

Point 35: It would be good to have more discussion in the background about other therapeutic HIV vaccines and how they compare with the candidate vaccine described in this manuscript. 

Response 35: the information has been added

Point 36: Line 122- should read “This study

Response 36: corrected

Point 37: Line 122- it is not clear what criteria were going to be used to select a dose.  In some ways this trial represents a Phase I dose-finding study more than a Phase II trial where the study was not designed to look at efficacy.

Response 37: In Phase I clinical trial no dose dependence for 1 mg was found. The results of Phase II suggest that we may decrease a dose below 0.25 mg.

Point 38: Line 129- this information belongs in the results, not the discussion.

Response 38: removed

Point 39: Line 132- before it can concluded that the vaccine is safe, more information is needed about the neutropenia- did it resolve?  How extensive was it? Why was it seen more in the lower dose vaccine group?

Response 39: Despite the neutropenias were defined as adverse events of 3rd and 4th grade it was not considered as life threatening by doctors and did not require the interruption of the vaccination. In all cases neutropenias were completely resolved. Paradoxically, the lower concentration of DNA vaccine (0.25 mg) caused more adverse events than 0.5 mg, which may be an indicator that lower dose is more active.

Point 40: Line 137- should read “Recently information..”

Response 40: corrected

Point 41: Line 138- should read “after a similar procedure..”

Response 41: corrected

Point 42: Line 145- should read “vaccines…”

Response 42: corrected

Point 43: Line 148- should read “as shown in a phase I…”

Response 43: corrected

Point 44: Line 149- should read “in a cohort”

Response 44: corrected

Point 45: Line 154- as the authors indicate, there are several possible explanations of blips appearing.  No real evidence that this was due to reactivation of the latent viral reservoir was presented.  As the authors indicated, neither relative frequency of the blips nor relative numbers of patients with blips differed between groups.  The only evidence presented were sizeable blips in 2 patients in the lower dose vaccine group, and a smaller one in one patient in the larger dose group.  Do we have information on consistent use of ART by these patients? Might there have been any other concurrent events that might have explained those blips, e.g., a cold or other infection?

Response 45: We did not have any information on other concurrent events. Although the number of sizable blips was small, all of them happened to be in blinded vaccinated groups.

The authors.

Reviewer 2 Report

This is a good report of a simple Phase II clinical trial of a putatively therapeutic DNA vaccine for use in HIV-infected people, which showed that the vaccine was safe and moderately well tolerated. I am a little concerned however that, if the authors claim that TNFAlpha can cause reservoir reactivation, that they didn't assess levels in patients. The level of HIV-specific CD8+ T-cells was also not reported, which I would have thought would be a necessary result for such a vaccine. At the least I would like to see discussion around this point?

Author Response

Dear Reviewer,

thank you for reviewing the manuscript. Please, find attached a point-by-point response to your comments.

Point 1: text corrections 

Response 1: all text corrections have been made.

Point 2: This is a good report of a simple Phase II clinical trial of a putatively therapeutic DNA vaccine for use in HIV-infected people, which showed that the vaccine was safe and moderately well tolerated. I am a little concerned however that, if the authors claim that TNFAlpha can cause reservoir reactivation, that they didn't assess levels in patients. The level of HIV-specific CD8+ T-cells was also not reported, which I would have thought would be a necessary result for such a vaccine. At the least I would like to see discussion around this point?

Response 2: Immunogenicity of the vaccine was studied in Phase I clinical trial by IFNγ-ELISpot, intracellular cytokine staining of IFNγ, TNFα, and IL-2, lymphocyte proliferation assay and ELISA. In Phase II clinical trial we assessed just common immunological parameters of trial participants to study the influence on the vaccination on immune system of patients. This is added to the "discussion" section.

In Phase I we found that DNA-4 vaccination induced increased TNFα expression in some individuals, so we hypothesized that viral blips observed in several participants of Phase II trial might be cause by TNFα what is consistent in the literature data. We agree that this assumption requires further study, and plan to perform this in our further studies.

The authors.

Reviewer 3 Report

General comments: Authors present the results of a Phase II clinical trial where they referred an evaluation of the safety, immunogenicity and dose selection of a DNA-based HIV therapeutic vaccine. Its an interesting approach but does not meet some of the primary endpoints referred.

Please have the manuscript review by a native English speaker for grammar and sentence structure.

Introduction: last sentence describe your results, please state the main objective of the study instead

Methods:

This section needs major improvement

Study vaccine --> Since the vaccines were prepared with a plasmid concentration of 0,25mg/ml, does this means that the vaccines dosing 0.5mg had more volume? 

Design --> there is no equal distribution in each group since placebo has 3 more participants, can you explain why this difference? How was the randomization performed?

Can you explained why and how were  chosen the referred plasmid concentrations? 

The evaluation of adverse events is well described. 

In the abstract immunogenicity is referred as a primary endpoint, how it was evaluated? Did you performed an elispot? CD4/CD8 ratio can not be considered as a proper immunogenicity evaluation

Results

This section needs major improvement

Consort diagram is recommended

Table 1 is confusing, data should be presented reversing the axis

About the patient withdrawal, can you define why the adverse event was unrelated?

Table 2 define ALT, GGT

If there were grade 3-4 neutropenia they should be considered as related to the vaccine and as a safety matter, should not this be a reason for stopping the vaccination? How did the investigators approach this situation? Are these adverse events solved?

Table 3 is confusing, could be synthesize

About described "blips" how did the investigators assessed the adherence to ARV? 18,000 copies/mL seems much more than a blip. Were virus resistances assessed?

Table 4 Data is in percentage? total number? 

Discussion

This section needs mayor improvement

Second paragraph of the discussion should be an overall review of the main results 

Second paragraph is confusing, description of more adverse events in the intervention arm it what could be expected, I don't see the relation with a small sample. 

Investigators could ponder more in the reasons for having same side effects on placebo arm and the lower dose vaccine. 

Describing changes in viral load as a marker of changes in the reservoir is an overstatement 

No limitations description

Needs an overall conclusion 

Author Response

Dear Reviewer,

thank you for reviewing the manuscript. Please, find attached a point-by-point response to your comments.

Point 1: Please have the manuscript review by a native English speaker for grammar and sentence structure.

Response 1: done

Point 2: Introduction: last sentence describe your results, please state the main objective of the study instead

Response 2: corrected

Point 3: Study vaccine --> Since the vaccines were prepared with a plasmid concentration of 0,25mg/ml, does this means that the vaccines dosing 0.5mg had more volume? 

Response 3: the information has been added

Point 4: Design --> there is no equal distribution in each group since placebo has 3 more participants, can you explain why this difference? How was the randomization performed?

Response 4: the information about randomization has been added. Due to financial default which happened in Russia during this period, the number of recruited participants was lower than planned.

Point 5: Can you explained why and how were  chosen the referred plasmid concentrations? 

Response 5: the information has been added

Point 6: The evaluation of adverse events is well described. 

Response 6: the information has been added

Point 7: In the abstract immunogenicity is referred as a primary endpoint, how it was evaluated? Did you performed an elispot? CD4/CD8 ratio can not be considered as a proper immunogenicity evaluation

Response 7: Immunogenicity of the vaccine was studied in Phase I clinical trial by IFNγ-ELISpot, intracellular cytokine staining of IFNγ, TNFα, and IL-2, lymphocyte proliferation assay and ELISA. In Phase II clinical trial we assessed just common immunological parameters of trial participants to study the influence on the vaccination on immune system of patients.

Point 8: Consort diagram is recommended

Response 8: the diagram has been added

Point 9: Table 1 is confusing, data should be presented reversing the axis

Response 9: corrected

Point 10: About the patient withdrawal, can you define why the adverse event was unrelated?

Response 10: the explanation has been added

Point 11: Table 2 define ALT, GGT

Response 11: done

Point 12: If there were grade 3-4 neutropenia they should be considered as related to the vaccine and as a safety matter, should not this be a reason for stopping the vaccination? How did the investigators approach this situation? Are these adverse events solved?

Response 12: Despite the neutropenias were defined as adverse events of 3rd and 4th grade it was not considered as life threatening by doctors and did not require the interruption of the vaccination. In all cases neutropenias were completely resolved. Clinicians considered neutropenias as a consequence of ART.

Point 13: Table 3 is confusing, could be synthesize

Response 13: done. New Table 3 has been generated. The previous table has been moved to Appendix A. The viral load at screening has been added to the Appendix Table in accordance with recommendation of the other reviewer.

Point 14: About described "blips" how did the investigators assessed the adherence to ARV? 18,000 copies/mL seems much more than a blip. Were virus resistances assessed?

Response 14:  The adherence was assessed by self-reported data. Virus resistance was not assessed.

Point 15: Discussion: This section needs mayor improvement. 

Response 15: done. The discussion of immunogenicity has been added. The second paragraph has been improved.

Point 16: Table 4 Data is in percentage? total number? 

Response 16: the information has been added

Point 17: Second paragraph of the discussion should be an overall review of the main results 

Response 17: done, the information has been added

Point 18: Second paragraph is confusing, description of more adverse events in the intervention arm it what could be expected, I don't see the relation with a small sample. 

Response 18: We agree with reviewer that wording was wrong. The corrections has been done.

Point 19: Investigators could ponder more in the reasons for having same side effects on placebo arm and the lower dose vaccine. 

Response 19: the frequency of local reactions in group immunized with 0.25 mg of the vaccine and the placebo group was similar, but the total number of adverse events in 0.25 mg group was twice higher than in Placebo group despite the fact that the size of Placebo group was larger. More discussion of these issue has been added.

Point 20: Describing changes in viral load as a marker of changes in the reservoir is an overstatement 

Response 20: Describing changes in viral load is modified.

Point 21: No limitations description

Response 21: The serious limitations of the study were caused by default of Russian currency (two-fold) right in the middle of recruitment process. This caused the reduction in the number of participants and in decrease of the immunogenicity studies. We concentrated on spontaneous increases of viral load studies because it seemed to us more scientifically interesting.

Point 22: Needs an overall conclusion 

Response 22: done.

The authors.

Round 2

Reviewer 1 Report

The authors have done a good job in responding to many of the comments and the manuscript  is better than it was.  The data indicating that the vaccine is showing evidence of possible efficacy are still weak, although the authors do a better job at discussing this in the revised manuscript.  They did not add any statistics to Table 2 as requested.

Author Response

Dear Reviewer,

thank you for reviewing the manuscript. Please, find attached the author's response to your comments.

Point 1: The data indicating that the vaccine is showing evidence of possible efficacy are still weak, although the authors do a better job at discussing this in the revised manuscript.   â€¨

Response 1: The authors agree with Reviewer that the data are weak, but they can help in planning future experiments, e.g. by recruiting participants with HLA molecules correlated with DNA vaccine efficacy, and/or increasing the delivery efficiency by electroporation.

Point 2: They did not add any statistics to Table 2 as requested.

Response 2: The statistic data have been added.

The authors.

Reviewer 2 Report

The ms is considerably changed since last I saw it, and much improved. Two things concern me, though, which are related, and which really do need to be discussed.

First, I see no mention of engineering the sequences of the vaccine genes to kill the biological activity of the expressed proteins. This is regarded as absolutely necessary in any consideration of Nef, Tat etc as vaccine candidates, as otherwise unwanted activities will occur - like transactivation of dormant HIV genomes. Was this tested?

Second, the "blips" could conceivably be due to the vaccine proteins activating latent genomes: was this tested in vitro, could it be a factor? This needs discussion before I could see this paper published.

Author Response

Dear Reviewer,

thank you for reviewing the manuscript. Please, find attached the author's response to your comments.

Point 1: First, I see no mention of engineering the sequences of the vaccine genes to kill the biological activity of the expressed proteins. This is regarded as absolutely necessary in any consideration of Nef, Tat etc as vaccine candidates, as otherwise unwanted activities will occur - like transactivation of dormant HIV genomes. Was this tested?   â€¨

Response 1: The information on engineering and modification of the sequences of vaccine genes has been added. No attempts to kill transactivation activities of Nef protein have been performed.

Point 2: Second, the "blips" could conceivably be due to the vaccine proteins activating latent genomes: was this tested in vitro, could it be a factor? This needs discussion before I could see this paper published.

Response 2: Done, the discussion of the possible involvement of Nef protein is added.

The authors.

Reviewer 3 Report

I appreciate very much all the changes to improve this manuscript.

I suggest some minor changes in order to accept it: 

Methods: Inclusion and exclusion criteria could be added as an annex.

Results: Table 4 use dots "." and commas "," indistinctly, please use only one. 

Discussion: Please summarize the results of phase 1 trial, even do gives the rationale for this study are 2 different ones. 

Author Response

Dear Reviewer,

thank you for reviewing the manuscript. Please, find attached the author's response to your comments.

The authors.

Round 3

Reviewer 1 Report

There remain some serious methodological limitations to this study but overall the manuscript is improved.  It remains unclear whether the HIV blips indicate true vaccine activity. 

Reviewer 2 Report

I thank the authors for addressing my concerns